# EXPLICIT-CONSTRAINED SINGLE AGENT FOR ENHANCED TASK-SOLVING IN LLMS

## ABSTRACT

In this study, we introduce the Explicitly Constrained Agent (EC-Agent), a novel approach designed to enhance the task-solving capabilities of Large Language Models (LLMs). Unlike existing multi-agent systems that depend on agents evaluating tasks from different perspectives, EC-Agent explicitly imposes task-oriented constraints for LLMs. Our observations are two-fold: first, assigning agents to sub-tasks with defined responsibilities implicitly sets constraints; second, these multi-agent systems often struggle with accurately assigning agents to sub-tasks, leading to overlapping duties and potential misguidance. In contrast, our single-agent system, driven by explicit methods and constraints, provides LLMs with detailed prompts, resulting in more precise responses. EC-Agent consists of two stages: a Reasoning Stage and a Summary Stage. 1) In the Reasoning Stage, three modules are proposed: Explicit Method, Explicit Constraint, and Execution. Specifically, LLMs utilize the Explicit Method and Constraint modules to analyze the task type and specific rules, generating multiple suitable methods and constraints. Subsequently, the Execution module combines these methods and constraints to produce and output possible solutions. 2) In the Summary Stage, LLMs evaluate the multiple reasoning processes and results from the previous step. They rectify any inconsistencies, summarize the information, and output the final result. Experimental results demonstrate that EC-Agent outperforms previous methods across a variety of tasks.

## 1 INTRODUCTION

The widespread applications of Large Language Models (LLMs) has led to transformative advances in various domains. However, the potential of LLMs is still under-exploited, especially for high-complexity tasks such as creative writing and high-level work plan design according to Qin et al. (2023), where the LLMs suffer a lot from issues related to the strength of knowledge and reasoning caused by problems such as hallucination (Bang et al., 2023; Bubeck et al., 2023), lack of slow thinking (Sloman, 1996; Lin et al., 2023a) and so on.

Recent research has attempted to address these challenges by introducing structured intermediate steps and leveraging the ability of multi-persona systems. The Chain of Thought (CoT) (Wei et al., 2023) approach improves LLM performance by breaking down the task into smaller, easier handle steps, leading to more thorough reasoning at each stage. Similarly, the Tree of Thought (ToT) (Yao et al., 2023a) approach uses a branching strategy to explore multiple potential solutions at the same time, thereby increasing the chances of finding the right and best result. In addition, the Self-Planner Prompt (SPP) (Wang et al., 2023b) study illustrates the effectiveness of dynamically allocating agents in the LLM based on specific task inputs and user-defined requirements. This approach enables the model to generate more rational and contextually appropriate outputs by adapting the reasoning process to the task's unique demands.

However, our observation is that multi-agent systems implicitly set constraints based on the task and rely on agents to evaluate the task from different perspectives. While these systems aim to improve instruction generation by introducing agents with distinct roles and responsibilities, they face practical challenges. Accurately generating executable plans for the LLM and allocating agents to specialized sub-tasks and defining their responsibilities can be difficult, potentially leading to overlapping duties. Furthermore, the effectiveness of these solutions depends on the roles of the generated agents and

Figure 1: The pipeline of EC-Agent comprises two stages: the Reasoning Stage and the Summary Stage. Key modules within this structure are the Explicit Method (EM) and Explicit Constraints (EC).

their operational sequencing, with a single miss-assigned or unrelated agent potentially misleading the entire reasoning process.

To address these issues, we propose the EC-Agent, a novel approach that leverages a single-agent system driven by explicit method and constraints. By providing LLMs with prompts that combine explicit method and constraints, our approach provides more accurate and efficient reasoning during task-solving.

To be more specific, EC-Agent involves two stages: the Reasoning Stage and the Summary Stage. The Reasoning Stage consists of three modules: the explicit method generation module (EM), the explicit constraints generation module (EC), and the execution module. The input task is first analyzed to infer a suitable method for solving it, along with explicit and implicit constraints, ensuring the model accurately identifies the necessary conditions for task solutions. Following this, EC-Agent guides the LLMs in combining the task description with the generated method and explicit constraints to formulate a comprehensive execution plan. This plan, which includes a detailed multi-step work strategy, significantly enhances the LLMs' reasoning accuracy. In the Summary Stage, EC-Agent reviews the outputs of the execution process. It prompts the LLMs to rectify and summarize the results based on the initial task description, method, and constraints, ensuring a high-quality final solution.

We further conduct experiments on various tasks, including 'Trivia Creative Writing,' 'Logic Grid Puzzles,' 'Codenames Collaborative,' and 'Code Generation.' EC-Agent improves the performance of LLMs compared with standard prompting and outperforms previous methods.

In summary, EC-Agent is an intelligent agent driven by explicit task constraints and planning. It effectively solves general tasks while avoiding some pitfalls associated with multi-role agent systems. Our main contributions are as follows:

1. EC-Agent combines the Reasoning Stage and the Summary Stage in a structured pipeline that is simple yet ensures thorough and systematic task-solving.

2. By incorporating explicit method and constraints, EC-Agent provides clear and direct guidance to LLMs, improving the accuracy and reliability of their outputs.

3. Experimental results show that EC-Agent improves LLM performance compared to standard prompting and outperforms previous methods with performance margins on various tasks.

## 2 RELATED WORK

### 2.1 PROMPTING PIPELINE

Wei et al. (2023) introduces the concept of Chain-of-Thought, which effectively enhances the reasoning ability of LLMs by generating a series of intermediate reasoning steps in response to a question. Yao et al. (2023a) proposes to explore multiple feasible paths and combine searching and backtracking, which improves the effectiveness in solving complex logical reasoning problems. Besta et al. (2023) introduces a GoT structure that can comprehensively utilize the optimal results generated

during the reasoning process. While ongoing advancements aim to augment the task-solving ability of LLMs through the implementation of diverse reasoning pipelines, a notable performance gap persists, particularly in generating complex content requisite for domains such as creative writing and storyboard generation. In this study, we explore a new pipeline imbued with a hierarchical structure specifically engineered to address the complexities inherent in such creative generation tasks. these methods have two limitations: 1) They cannot generate complex content. 2) The scale of problems they can solve is relatively limited.

## 2.2 AUTONOMOUS AGENTS FRAMEWORK

LLMs have emerged as fundamental components within the autonomous agents framework. This framework cover three distinct operational modes: single-agent mode, human-agent interaction mode, and multi-agent mode. Gravitas (2023) is an early single-agents framework that relies on LLMs, combined with relevant tools, to address issues of diversity. Hong et al. (2023), Cai et al. (2023), Lin et al. (2023b) are two human-agent interaction frameworks that enhance task performance through continuous human-machine interaction. Nakajima (2023) is composed of multiple agents, each with a specific role: the task creation agent generate task, the task prioritization agent manages the task list, and the execution agent selects and executes tasks from this list. The system operates with fixed agents and adheres to a predetermined sequence of execution. Hong et al. (2023) has shown remarkable advancements in Agent-Pipeline solutions, demonstrating unparalleled creativity and proficiency in problem resolution. Li et al. (2023) introduce systems optimizing inter-agent communication. Park et al. (2023) builds on a multi-agent foundation, providing the capability for self-refine based on results. Wu et al. (2023) allows users to create and manage multiple autonomous agents to collaboratively complete complex tasks. Wang et al. (2023b) proposes a reasoning method with automatic evaluation. This approach involves automating the setup of multiple agents with different ability to evaluate and improve reasoning results multiple times. It endows LLMs with stronger reasoning abilities while effectively reducing hallucinations. Zhou et al. (2024) incorporation of an environment for external feedback. However, these methods depend on the accuracy of the LLM in generating agents. Hallucinations by the LLM, such as creating mismatched agents or making unreasonable task arrangements, can negatively impact the effectiveness of task execution.

## 2.3 SPECIFIC TASKS-SOLVING WITH AGENT

In ths field of trivia creative writing, Mirowski et al. (2023) have employed hierarchical and structured approaches combined with specific role assignments to leverage the ability of LLMs in generating long-form creative content, such as continuous scripts enriched with detailed contextual elements. These studies, along with Zhang et al. (2019), Mishra et al. (2023), Liu et al. (2023a), have made commendable strides in integrating domain-specific knowledge to guide LLMs in executing tasks with enhanced precision in respective domains.

The series of Chain of Thought (CoT) articles, such as Diao et al. (2023), Shum et al. (2024), Wang et al. (2023a), demonstrate various reasoning and planning generation methods. Le et al. (2024) enhances the reasoning ability of LLMs by combining language models with code execution in a form similar to CoT. Huang et al. (2024) builds on this CoT approach by incorporating the concept of test cases. Shinn et al. (2023) improves the execution of logic generation and planning tasks by emulating human reflection. Lin et al. (2024) assigns LLM agents specific roles in typical development teams to improve code generation quality.

Despite the plethora of advancements and innovations in LLMs, a common limitation is evident, i.e., the reliance on manual specification for both problem-solving processes and the determination of agent attributes, impeding their versatility in generalized applications.

## 2.4 COGNITIVE SCIENCE AND LLMS

Researchers such as Piaget (2013), Pellegrini (2009), Wason & Johnson-Laird (1972), Sloman (1996) have researched into the realms of human psychology and cognition, influencing subsequent developments in artificial intelligence theory (Chandrasekaran et al., 2017). Devlin et al. (2019), Brown et al. (2020), OpenAI (2023), Chowdhery et al. (2022), Srivastava et al. (2023) demonstrate the ability of large language models (LLMs) to the public. Shuster et al. (2022), Bang et al. (2023),

Liu et al. (2023b), Yao et al. (2023b), Lin et al. (2023a), Madaan et al. (2023), Shinn et al. (2023), by integrating cognitive science with Large Language Models (LLMs), continuously explore methods to enhance the ability of LLMs. Research related to cognitive load theory (September, 2018), indicates that when teachers instruct students on solving new problems, clear solution design, practice, and feedback are more effective for problem resolution than requiring students to independently explore all aspects they need to learn for solving the problem.

All of these approaches represent new explorations in both theoretical and methodological aspects, providing fresh perspectives for the enhancement and development of LLMs in the future.

## 3 METHOD

In this section, we formally introduce EC-Agent, which consists of two stages: Reasoning and Summary, as demonstrated in Figure 1. We begin by revisiting the definition of task-solving for LLM agents, followed by a detailed explanation of EC-Agent, including its key components and the complete pipeline.

Given an input instruction $x$ and a model $\mathcal{M}$, if we denote the final output to be $y$, then the Standard Prompting (Equ. 1) and Chain-Of-Thought (Equ. 2) can be formulated as:

$$y = \mathcal{M}(x) \tag{1}$$

$$y = \mathcal{M}(p_{cot}|x|\{z_1, z_2, .., z_n\}) \tag{2}$$

where $p_{cot}$ is the CoT prompt and $z_1, z_2, ...z_n$ are the implicit intermediate steps.

In contrast, our EC-Agent can be divided into two stages: 'Reasoning' and 'Summary.' In the 'Reasoning' stage, EC-Agent aims to identify and generate suitable explicit methods and explicit constraints from user input $x$. Specifically, it generates multiple methods suitable for solving the task, represented as $\mathcal{F}(x) = (f_1, f_2, f_3, ...)$, along with a set of constraints $\mathcal{C}(x) = (c_1, c_2, c_3, ...)$, all based on user input $x$. Next, results are generated based on $\mathcal{F}(x)$ and $\mathcal{C}(x)$, denoted as $\mathcal{E}(f, c) = (e_1, e_2, e_3, ...e_n)$, where $(e_1, e_2, e_3, ...e_n)$ represents the implicit execution sequence. In the 'Summary' stage, it includes both a review section and a summary section. The review section, denoted as $\mathcal{A}(e, c) = (a_1, a_2, a_3, ...a_n)$, is used to examine and adjust the reasoning process as well as the validity of the results. The summary section, denoted as $\mathcal{S}(a, c)$, integrates all reasoning processes and answers, and based on the task and constraints, generates the final result. The system can be represented as:

$$y = \mathcal{M}(p_{ec}|x|\mathcal{F}(x)|C(x)|\mathcal{E}(f, c)|\mathcal{A}(e, c)|\mathcal{S}(a, c)) \tag{3}$$

We now provide details for each stage of EC-Agent. Detailed prompts of each module can be found in Appendix A.

### 3.1 REASONING STAGE

The first stage of EC-Agent is reasoning, where it can generate multiple possible solutions given user input $x$. At the core of this stage are the creation of methods and constraints with respect to the tasks. This is then followed by an execution step to develop the solutions. We now elaborate on the detailed functionality of the three main components in reasoning: Explicit Method Generation, Explicit Constraints Generation, and Execution.

### 3.1.1 EXPLICIT METHOD GENERATION

A core module of EC-Agent is the generation of explicit methods. In the first step of the reasoning stage, EC-Agent guides the LLM to search for and generate one or more methods suitable for the given task based on the input task information. These methods are then output in an explicit form as execution plans. During this phase, we prompt the LLM to generate general methods rather than those specifically tailored to the task requirements. This process is represented as $\mathcal{F}(x)$, and the generated results are denoted as $(f_1, f_2, f_3, ...)$.

### 3.1.2 EXPLICIT CONSTRAINTS GENERATION

Another core module of EC-Agent is the generation of explicit constraints. Previous work has shown that multi-agent systems benefit from evaluating outputs based on the roles of the agents. We posit that providing constraints to LLMs during problem-solving can achieve similar performance enhancements, with explicit constraints offering direct and efficient guidance. Specifically, these explicit constraints are derived from two aspects:

1. Explicit Requirements from the Input Task: This includes specific instructions embedded within the task, such as "5 or 10 common-sense questions in Trivia Creative Writing tasks" or "room numbers with person preferences in Logic Grid Puzzle tasks."

2. Implicit Constraints Related to the Task: This includes inherent requirements necessary for the task's execution, such as "validity checks for input values in code generation tasks."

This process is represented as $\mathcal{C}(x)$, and its output results are as $(c_1, c_2, c_3, ...)$.

### 3.1.3 EXECUTION

Based on the generated $\mathcal{F}(x)$ and $\mathcal{C}(x)$, EC-Agent further guides the LLMs to perform reasoning execution. This involves synthesizing the current task description with the generated methods and constraints, and then outputting the reasoning results, denote as $\mathcal{E}(f, c)$.

To enhance the robustness of EC-Agent, we employ a multi-sampling approach. This approach allows the LLMs to execute the Reasoning Stage multiple times, generating several results for the task. This increases the likelihood of producing feasible and accurate solutions. To maintain a balance between efficiency and cost, we use 2 samples in our experiments.

## 3.2 SUMMARY STAGE

Following the execution results from the Reasoning stage, the Summary stage is employed to produce the final solution. This stage consists of two main steps: Review and Conclusion. In the Review step, each execution plan's results are evaluated independently to ensure correctness, consistency, and adherence to the constraints. In the Conclusion step, the LLMs synthesize all solutions and their evaluations, summarizing the outcomes into a coherent and accurate final result.

### 3.2.1 REVIEW

In this step, a list of 'common errors for the task' is generated by combining the 'task,' 'method,' and 'constraints.' This is followed by a reflection and correction of the input "reasoning information and results," and the corrected "reasoning process and results" are re-output, denoted as $\mathcal{A}(e, c)$.

### 3.2.2 CONCLUSION

The second step is 'Conclusion,' denoted as $\mathcal{S}(a, c)$. We instruct the LLMs to summarize and select based on the task description and all reasoning records (each containing a complete reasoning process and outcome), discarding alternative answers with flawed logical reasoning, ultimately providing the final result. Throughout this process, we observe that the LLMs pay more attention to differences and the logic in the reasoning process. Based on this information, they summarize and select, leading to a performance improvement.

## 4 EXPERIMENTS

To assess the efficacy and adaptability of EC-Agent across different applications, we apply EC-Agent to 4 distinct tasks, including **Trivia Creative Writing**, **Logic Grid Puzzle**, **Codenames Collaborative** and **Code Generation**.

| Methods | Trivia.C.W (N=5) | | Trivia.C.W (N=10) | | Logic.G.Puzzle | | Codenames.C | |
|---|---|---|---|---|---|---|---|---|
| (GPT-4-0613) | Score (%) | Δ | Score (%) | Δ | Score (%) | Δ | Score (%) | Δ |
| Standard Prompting | 74.6 | | 77.0 | | 57.7 | | 75.4 | |
| CoT | 67.1 | ↓7.5 | 68.5 | ↓8.5 | 65.8 | ↑8.1 | 72.7 | ↓3.3 |
| Self-Refine[iter=1] | 73.9 | ↓0.7 | 76.9 | ↓0.1 | 60.0 | ↑2.3 | 64.6 | ↓10.8 |
| SPP | 79.9 | ↑5.3 | 84.7 | ↑7.7 | 68.3 | ↑10.6 | 79.0 | ↑3.6 |
| **EC-Agent (ours)** | **80.5** | ↑5.9 | **85.7** | ↑8.7 | **69.2** | ↑11.5 | **82.9** | ↑7.5 |

Table 1: We compared with Standard Prompting, CoT, Self-Refine, and SPP. EC-Agent outperforms these baselines in various tasks. (Based on GPT-4-0613)

| Methods | Trivia.C.W (avg) | | Logic.G.Puzzle | | Codenames.C | |
|---|---|---|---|---|---|---|
| (GPT-3.5-Turbo) | Score (%) | Δ | Score (%) | Δ | Score (%) | Δ |
| Standard Prompting | 64.0 | | 46.0 | | 61.0 | |
| CoT | 48.0 | ↓16.0 | 50.0 | ↑4.0 | 54.0 | ↓7.0 |
| SPP | 37.0 | ↓27.0 | 48.0 | ↑2.0 | 35.0 | ↓26.0 |
| **EC-Agent (ours)** | **71.4** | ↑7.4 | **50.0** | ↑4.0 | **66.8** | ↑5.8 |

Table 2: EC-Agent demonstrated a certain level of stability, without experiencing the performance instability as in CoT or SPP. (Based on GPT-3.5-Turbo-0613)

| Model | Methods | Trivia.C.W (N=5) | | Trivia.C.W (N=10) | | Logic.G.Puzzle | | Codenames.C | |
|---|---|---|---|---|---|---|---|---|---|
| | | Score (%) | Δ | Score (%) | Δ | Score (%) | Δ | Score (%) | Δ |
| LLama3.1-8b | Standard Prompting | 50.4 | | 60.1 | | 44.0 | | 50.1 | |
| | **EC-Agent (ours)** | **59.4** | ↑9.0 | **65.9** | ↑5.8 | **48.3** | ↑4.3 | **62.3** | ↑12.2 |
| Gemma2-9b | Standard Prompting | 52.6 | | 59.1 | | 43.5 | | 65.7 | |
| | **EC-Agent (ours)** | **58.9** | ↑6.3 | **63.2** | ↑4.1 | **51.0** | ↑7.5 | **75.3** | ↑9.6 |
| Mistral-7b | Standard Prompting | 46.4 | | 50.4 | | 33.5 | | 62.8 | |
| | **EC-Agent (ours)** | **49.4** | ↑3.0 | **52.8** | ↑2.4 | **42.5** | ↑9.0 | **67.5** | ↑4.7 |

Table 3: EC-Agent also achieves performance improvements across various tasks with different open-source models. More details about average and variance scores are provided in Appendix C.

## 4.1 TRIVIA CREATIVE WRITING

**Task Description.** We adopted the same task setting as SPP (Wang et al., 2023b) , requiring LLMs to craft coherent stories around a given topic while integrating answers to N trivia questions. This tests the model's capability to retrieve information from its self-compressed knowledge base. The task is structured into two scenarios: N=5 and N=10, where a higher N indicates a greater number of trivia questions to be correctly answered during story generation. SPP developed a benchmark comprising 100 instances for each scenario, utilizing a total of 1000 questions from the TriviaQA dataset (Joshi et al., 2017). We utilized the SPP dataset for both N=5 and N=10 settings in our tests.

**Evaluation Metrics.** We adopted the evaluation approach used by SPP, applying an automated metric to determine the ratio of target answers (and their variants) included in the generated output relative to the total number of trivia questions. This is quantified as the Trivia Creative Writing Metric Score, calculated by dividing the number of correct answer mentions by the total number of trivia questions.

**Result.** Table 1, Colume 1 & 2, Table 2, Colume 2, Table 3, Colume 3 & 4 demonstrate the exceptional performance of EC-Agent in model internal imformation retrieve. Compared to the standard method, CoT method, and generative Multi-Agent systems like SPP and Auto-Agent, EC-Agent achieved average scores of 83.1 (GPT-4-0613), 71.4 (GPT-3.5-Turbo-0613), 62.65 (LLama3.1-8b), 61.5 (Gemma2-9b), and 51.1 (Mistral-7b).

The inclusion of "explicit conditional constraint generation" in our method enhances the reasoning stability of the LLMs when answering factual questions, effectively reducing model hallucinations. With GPT-4-0613, our average scores are 85.7 for tasks with 10 trivia questions (N=10) and 80.5 for tasks with 5 trivia questions (N=5).

Compared to the Chain of Thought (CoT) and SPP methods, our approach provides better reasoning stability. Specifically, with the GPT-3.5-Turbo-0613 version, our method shows a 11.5% improvement over the Standard method and a substantial 92.9% improvement over the SPP method.

## 4.2 Logic Grid Puzzle

**Task Description.** In this task, we evaluate the multi-step reasoning ability of EC-Agent using the Logic Grid Puzzle task from the Bigbench dataset (Srivastava et al., 2023). This dataset comprises 200 instances, each describing a logic puzzle. The objective of each puzzle is to deduce house numbers by integrating given problems and clues, which detail 2-5 rooms, the residents, and specific characteristics of these residents.

**Evaluation Metrics.** For evaluation metrics, we assess the accuracy of the responses by comparing the predicted house numbers against the standard answers provided in the dataset.

**Result.** Table 1, Column 4 and Table 3, Column 4 present the results for the Logic Grid Puzzle task. EC-Agent enhances the reasoning and conditional constraint ability of large language models. Our evaluations show an average score of 69.2 (GPT-4-0613), 50.0 (GPT-3.5-Turbo-0613), 48.3 (LLama3.1-8b), 51.0 (Gemma2-9b), and 42.5 (Mistral-7b). These results demonstrate performance comparable to CoT and SPP, while exceeding both standard methods and Self-Refine (Madaan et al., 2023). (More detailed are provided in Task 2.1 of Appendix B)

## 4.3 Codenames Collaborative

**Task Description.** Codenames Collaborative is a collaborative task where two players take on the roles of Spymaster and Guesser. This task test the model's knowledge, reasoning, and theory of mind abilities. The Spymaster provides a hint word related to the target words while excluding other distractor words, and the Guesser uses the given hint and the full list of words to infer the target words.

**Evaluation Metrics.** Our task adopts the Codenames Collaborative dataset designed in SPP. This dataset provides an environment with a set of questions and target words, allowing for accurate measurement of the model's capability without human annotation.

**Result.** Table 1, Column 5 and Table 3, Column 5 present the results for the Codenames Collaborative task. Our evaluations show that EC-Agent achieves an average score of 82.9 (GPT-4-0613), 66.8 (GPT-3.5-Turbo-0613), 62.3 (LLama3.1-8b), 75.3 (Gemma2-9b), and 67.5 (Mistral-7b). This task demands more robust reasoning ability from LLMs compared to the trivia creative writing task. Notably, our method outperforms others, demonstrating superior knowledge and reasoning abilities in this collaborative setting. Please refer to Task 2.2 in Appendix B for more details.

## 4.4 Code Generation

**Task Description.** In this task, we evaluate the effectiveness of EC-Agent using code generation datasets: MBPP (Austin et al., 2021). The datasets consist of a comprehensive collection of Python programming problems designed to test a model's ability to handle various coding scenarios.

**Evaluation Metrics.** We use pass@1 as the evaluation metric for code correctness, the most widely adopted metric for automatic code generation (Chen et al., 2021; Huang et al., 2024).

**Result.** Table 4 presents the results of our evaluations using GPT-4-0613 and GPT-3.5-Turbo-1106 on the MBPP datasets, along with general task frameworks like Zero-Shot Prompting and MetaGPT, as well as specialized programming frameworks such as Language Agent T.S. and LLM CodeGen Scrum. Additionally, Table 5 showcases the results of our evaluations using open-source models on the same MBPP datasets.

In the basic MBPP, EC-Agent achieved average scores of 90.3 (GPT-4-0613), 83.2 (GPT-3.5-Turbo-1106), 72.8 (LLama3.1-8b), 72.4 (Gemma2-9b), and 49.5 (Mistral-7b). Our method, which

| Model | Methods | MBPP | |
|---|---|---|---|
| | | Score (%) | Δ |
| GPT-4 | Zero-shot Prompting | 82.5 | |
| | Meta GPT | 87.7 | ↑5.2 |
| | **EC-Agent (ours)** | **90.3** | ↑6.8 |
| GPT-3.5-Turbo | Zero-shot Prompting | 77.5 | |
| | Language Agent T.S. | 81.1 | ↑4.6 |
| | LLMCodeGen | 82.5 | ↑5.0 |
| | **EC-Agent (ours)** | **83.2** | ↑5.7 |

Table 4: Comparison of EC-Agent, baselines, and other methods across different models.

| Model | Methods | MBPP | |
|---|---|---|---|
| | | Score (%) | Δ |
| LLama3.1-8b | Zero-shot Prompting | 68.0 | |
| | **EC-Agent (ours)** | **72.8** | ↑4.8 |
| Gemma2-9b | Zero-shot Prompting | 69.1 | |
| | **EC-Agent (ours)** | **72.4** | ↑3.3 |
| Mistral-7b | Zero-shot Prompting | 49.0 | |
| | **EC-Agent (ours)** | **49.5** | ↑0.5 |

Table 5: MBPP test with open-source models.

emphasizes the explicit articulation of implicit conditions, improves model performance by ensuring the rationality of input data and generating test cases that follow the task's constraint instructions.

## 4.5 ABLATION STUDIES

We further conducted ablation studies on different modules of EC-Agent to evaluate the impact of each component on overall performance. The results of these evaluations are presented in Table 6.

**Explicit Method.** As shown in "EM" row in Table 6, Guiding the LLM to apply explicit methods in the reasoning process can efficiently enhances outcomes. This aligns with human cognitive processes, where a clear understanding of the task and selecting appropriate methods are crucial for successful completion.

**Explicit Constraints.** In our approach, we enhance the performance of LLMs by replacing the generation of dynamic role agents with explicit conditional constraints. The impact of the EC module on the results can be seen in the row titled 'EC' in Table 6. We also observer that the integration of explicit conditional constraints offers key advantages:

1. It eliminates the uncertainty and incompleteness in dynamic role generation.

2. It directs LLMs to focus more precisely on the task during the reasoning process. (Refer to Task 3 in Appendix B and Table 6)

**Multi Samples.** Additionally, we also tested the hypothesis: "Can the performance of problem-solving be improved by allowing the LLM to perform task planning multiple times under limited model ability?" by using a Multi Samples strategy. The results showed that while the model's reasoning performance benefits from additional repetitions of task planning, the associated reasoning cost also increases. (Refer to 'EM + EC + Samples' row in Table 6).

**Result Review.** Finally, we also evaluated the effects of 'self-summarization and reflection' on single and multiple reasoning results across various tests. The results indicated that the effects of 'summarization and reflection' varied across different tasks, depending on the method used to determine task outcomes. When task outcomes can be assessed through strict logical reasoning, this approach can enhance reasoning results; conversely, its effectiveness is weaker when the assessment method is less rigorous. (Refer to 'EM + EC + Review' row in Table 6).

| Methods (Gemma2-9b) | Trivia.C.W (N=5) | Trivia.C.W (N=10) | Logic.G.Puzzle | Codenames.C |
|---|---|---|---|---|
| Standard | 52.6 | 59.1 | 43.5 | 65.7 |
| Plan + CoT | 56.9 | 61.6 | 45.0 | 69.2 |
| Plan + CoT + EC | 55.6 | 62.1 | 45.5 | 69.3 |
| EM | 55.4 | 60.8 | 48.5 | 73.6 |
| EC | 56.4 | 61.3 | 47.0 | 74.0 |
| **EM + EC** | **57.0** | **61.6** | **50.0** | **75.1** |
| EM + EC + Review | 57.1 | 61.8 | 50.0 | 75.1 |
| EM + EC + Samples | 58.3 | 62.9 | 50.5 | 75.2 |
| **EC-Agent** | **58.9** | **63.2** | **51.0** | **75.3** |

Table 6: Ablation Studies: Comparative Analysis of Task Performance Scores for Various Methods. The experiments lead to the following conclusion: The EM + EC combination demonstrates stable performance across different task types, showing a slight advantage over the Plan + CoT combination in text generation tasks, and surpassing the Plan + CoT combination in reasoning tasks.

| Methods (Gemma2-9b) | Trivia.C.W (Avg.) | | Logic.G.Puzzle | | Codenames.C | |
|---|---|---|---|---|---|---|
| | input | output | input | output | input | output |
| Standard | 0.2k | 0.45k | 0.3k | 0.45k | 0.1k | 0.002k |
| EM+EC | 0.2k | 0.5k | 0.3k | 0.5k | 0.1k | 0.4k |
| EM+EC+Review | 0.7k | 0.9k | 0.8k | 0.9k | 0.5k | 0.8k |
| EM+EC+Samples | 2.1k | 1.9k | 2.4k | 1.5k | 1.5k | 1.2k |
| EC-Agent | 2.6k | 3.2k | 2.9k | 2.7k | 2.0k | 2.4k |

Table 7: Ablation Studies: Token (number) Costs Across Different Methods. As more modules are adopted, the consumption of tokens also increases. This is mainly due to multiple samples in the Reasoning Stage and reviews and conclusions in the Summary Stage.

**Computational Cost.** The cost of EC-Agent mainly comes from the multi-sample process in the Reasoning stage and the Summary stage. According to the ablation experiments, in tasks related to 'text content generation,' multi-sampling can effectively improve the success rate of the task. In 'logical reasoning' tasks, the impact of reducing the number of samples on the success rate is not as significant. Therefore, in practical applications, the number of samples can be adjusted appropriately based on the differences in task types to achieve a reasonable comparison between performance and cost. (Refer to Table 7).

**Alternate Modules.** We also conducted experiments with multiple module combinations. In this comparative experiment, we used 'Plan + CoT' as one module, combined with the EC module. The results of these experiments are shown in Table 6 under 'Plan + CoT,' 'Plan + CoT + EC,' and 'EM + EC' This cross-combination approach in multi-task experiments allows for a relatively objective validation of the effectiveness and stability of the system. The specific differences of these modules during the reasoning process can be referenced in Task 3 of Appendix B.

The following conclusions can be drawn from the experiments: for various tasks, while the combination of 'Plan + CoT' also achieved good results, its performance in some reasoning tasks was inferior to that of the 'EM + EC' combination ('EM + EC' is more focused on the specific requirements of the tasks). Therefore, the 'EM + EC' combination offers the better cost-effectiveness (including support for task diversity, reasoning results, reasoning efficiency, and computational cost). If the better performance is needed, running the complete EC-Agent Pipeline is recommended.

## 5 CONCLUSION

In conclusion, our study demonstrates that integrating explicit conditional constraints enhances LLMs performance by aligning model reasoning more closely with structured task execution method. Our findings also suggest that integration of explicit constraints and optimized task planning could pave the way for more robust and efficient LLMs applications in complex problem-solving scenarios.

LIMITATIONS

While EC-Agent demonstrates promising performance across various tasks, potential issues can arise in complex real-world scenarios. For instance, in extremely long reasoning chains, problems like 'early-termination' can occur. In the future, we will explore how to address reasoning problems in more complex scenarios and delve into the model's capabilities, particularly for open-source models.

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
