# EXPLICIT-CONSTRAINED SINGLE AGENT FOR ENHANCED TASK-SOLVING IN LLMS

# APPENDIX

## APPENDIX A

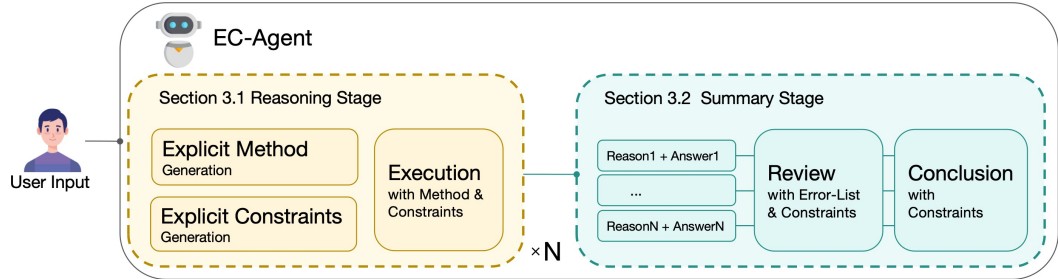

Figure 1: The pipeline of EC-Agent.

**EC-Agent Prompt:**

### Reasoning Stage Prompt:

{Task}
Step 1: Find a method.
Step 2: List and analyze all the constraints.
Step 3: Reason step by step, output the answer in the format: "Final Answer is:"

### Summary Stage Prompt:

### Review Module:

{Potential Answer}
{Task}
Step 1: List common errors in this type of task.
Step 2: Evaluate and Self-Reflection.
Step 3: Reason and output modified result.

### Conclusion Module:

{Potential Answer List}
{Task}
Step 1: Get result from potential answer, list and summaryk.
Step 2: Make a major vote on all potential answers.
Step 3: If all answers are incomplete, reason out a new one.
Step 4: Output the final result.

APPENDIX B

TASK 1: 9.8 OR 9.11, WHICH ONE IS BIGGER?

9.8 or 9.11, which one is bigger?

To determine which number is bigger, you simply compare them:

- **9.8** is less than **9.11**.

Thus, **9.11** is the larger number.

Figure 2: Reasoning by default prompt (GPT-4o).

9.8 or 9.11, which one is bigger?
Step1: Find a Method.
Step2: List and analyze all the constraints.
Step3: Reason, Output the answer. Format:"Final Answer is : "

**Step 1: Find a Method.**
To compare two decimal numbers, we can look at each digit from left to right.

**Step 2: List and analyze all the constraints.**

- Both numbers have the same number of digits before the decimal point (both are two-digit integers).

- The digits before the decimal point are the same for both numbers: '9'.

- We compare the digits after the decimal point (the tenths place): '8' for 9.8 and '11' for 9.11.

**Step 3: Reason, Output the answer.**
Since 9.11 has more digits after the decimal point, we need to consider that 9.11 can be expressed as 9.110. Thus, when comparing:

- 9.8 = 9.800

- 9.11 = 9.110

Now we compare:

- 9.800 (for 9.8)

- 9.110 (for 9.11)

Since 9.800 is greater than 9.110, we conclude:

**Final Answer is: 9.8**

Figure 3: Reasoning by EC-Agent (GPT-4o).

Correct reasoning method and conditional judgments can quickly guide the model to reach accurate conclusions.

## TASK 2: EC-AGENT VS STANDARD PROMPT

Compared to the Standard Prompt, EC-Agent, through the control of Explicit Constraints and Explicit Method., enables the LLM to not only generate diverse execution methods and drive subsequent reasoning behaviors of the LLM, but also focuses on the requirements of the Task during the task execution process.

### Task 2.1: Logic Grid Puzzle

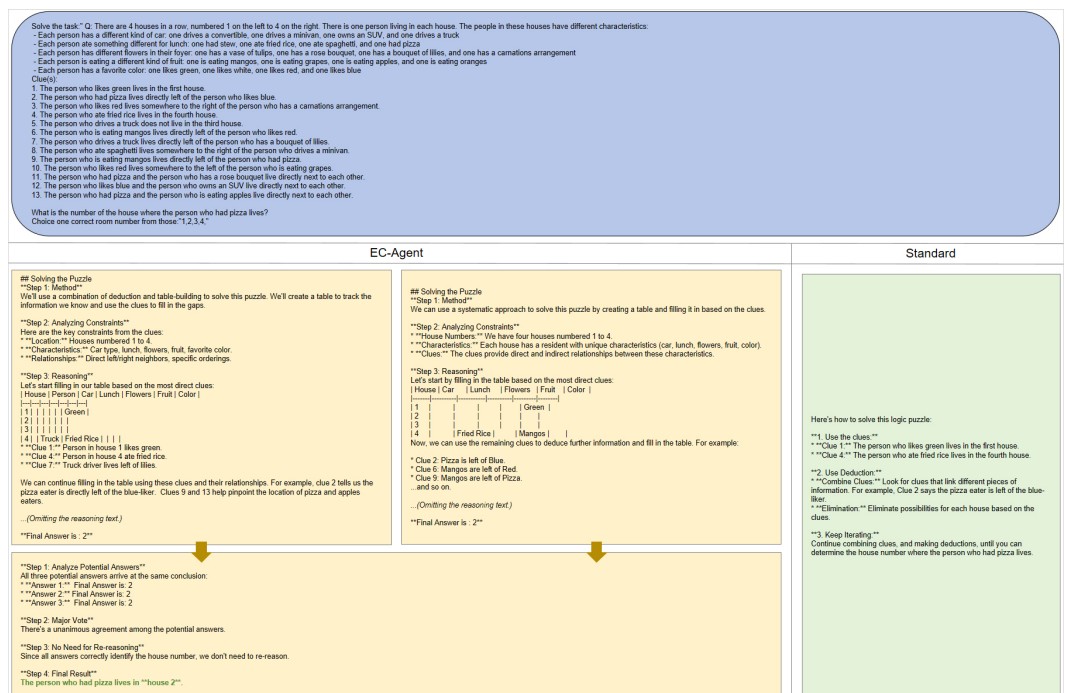

Figure 4: Logic Grid Puzzle Task. The Blue Grid is a task query. The yellow Grid is reason result by EC-Agent, The green Grid is reason result by Standard Prompt.

In Figure4, the "EC-Agent" selected a method of reasoning that combines tables with conditional constraints, based on the requirements of the Task. In the Puzzle task, we compared and analyzed the failure reasons of tasks using the "Standard Prompt" and "EC-Agent". In "EC-Agent", the occurrence of early-termination in reasoning decreased by 60% compared to Standard Prompt (from 20 cases to 8 out of 100 tasks). We believe this is because, after applying the EM&EC instructions, the LLM's reasoning process became more focused.

Task 2.2: Codenames Collaborative

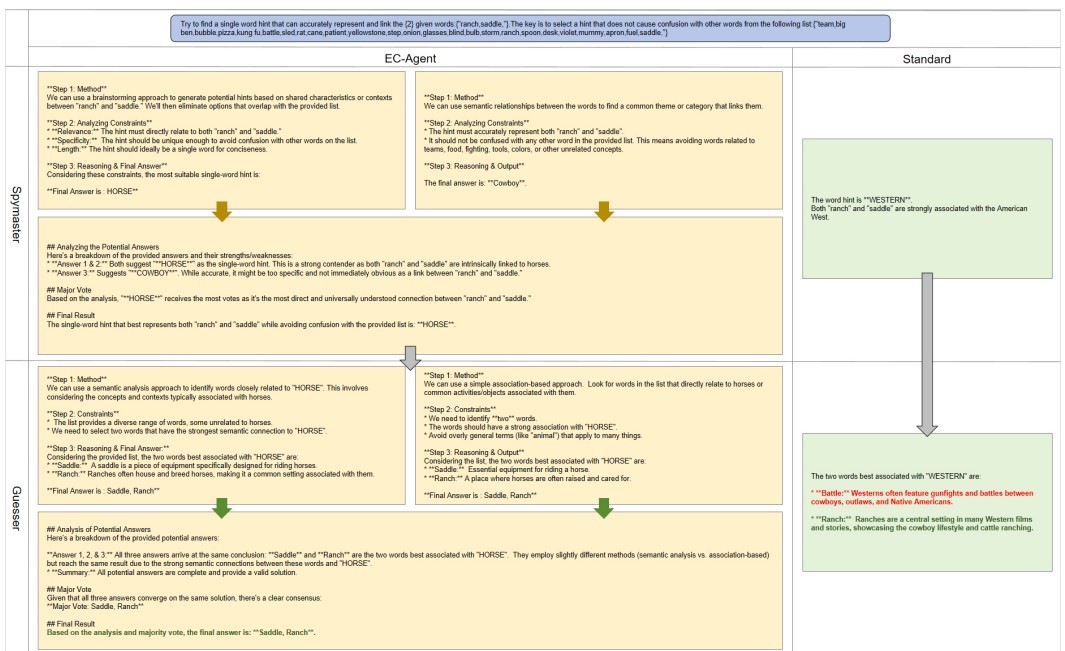

Figure 5: Codenames Collaborative. The Blue Grid is a task query. The yellow Grid is reason result by EC-Agent, The green Grid is reason result by Standard Prompt.

In Task 2.2, EC-Agent can specifically propose constraint conditions based on the Task requirements: "The hint should be unique enough to avoid confusion with other words on the list." And it follows this requirement during the reasoning and generation process: "Considering these constraints, the most suitable single-word hint is...". In Summary Stage, EC-Agent makes the correct choice for different "Potential Answers" by combining the requirements of the Task. In contrast, during the execution of the Standard Prompt, the LLM only considers the interrelationship between the two words "ranch" and "saddle" as required by the Task, ignoring the constraint relationships with other words, which leads to the failure of the final reasoning.

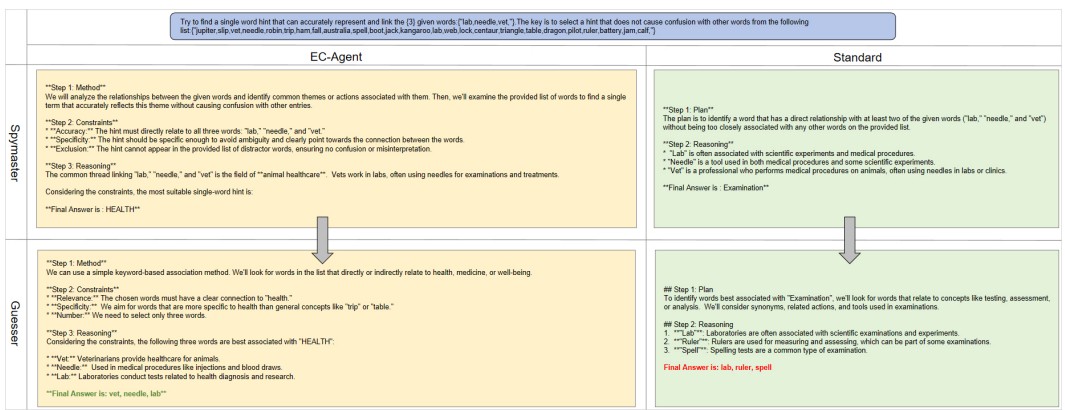

Figure 6: "Explicit Method+Explicit Constraints" vs "Plan". The Blue Grid is a task query. The yellow Grid is reason result by EC-Agent, The green Grid is reason result by "Plan" prompt.

**EM+EC (left):**

> {Task}
> Step 1: Find a method.
> Step 2: List and analyze all the constraints.
> Step 3: Reason step by step, output the answer in the format: "Final Answer is:"

**Plan+CoT (right):**

> {Task}
> Step 1: Make a plan.
> Step 2: Reason step by step, output the answer in the format: "Final Answer is:"

We tested the effect of replacing the prompt keyword "EM&EC" with "Plan," and observed a decline in evaluation results. Our analysis suggests that this is because using the "EM&EC" combination allows the LLM to focus more on the core objectives of the task while reducing reliance on the correctness of the plan. The table below records the test results using different prompts.

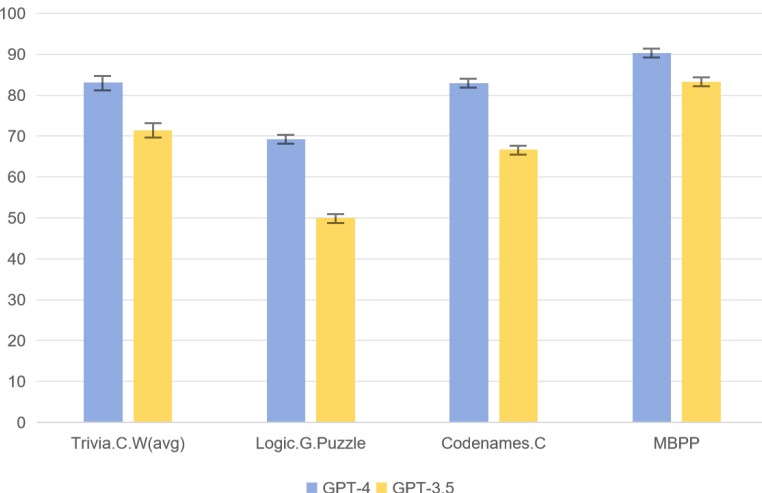

Figure 7: The scores and score variances of different tasks performed by the EC-Agent on OpenAI GPT series models.

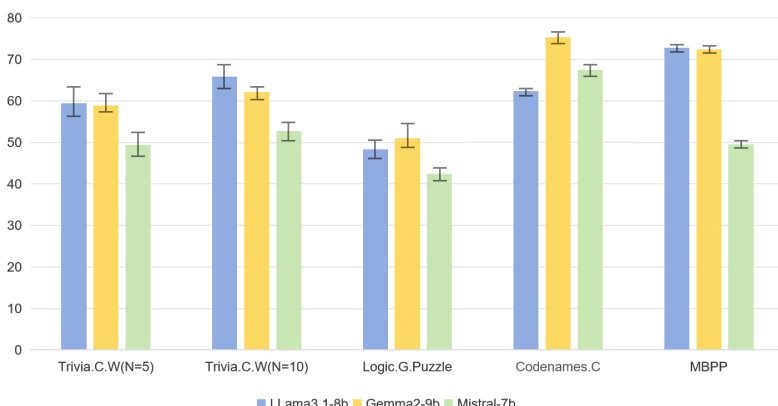

Figure 8: The scores and score variances of different tasks performed by the EC-Agent on multiple open-source models.

Combining numerical analysis, within the EC-Agent framework, some open-source models have reached or slightly surpassed the OpenAI GPT-3.5-0613 model on certain reasoning tasks (e.g., Logic G. Puzzle, Codename C.). However, there remains a significant gap to GPT-4-0613 overall. Additionally, different models also show variations in reasoning stability.