# OpenReview forum: "Explicit-Constrained Single Agent for Enhanced Task-Solving in LLMs"
_ICLR.cc/2025/Conference — ICLR 2025 Conference Withdrawn Submission_

### Official Review · Reviewer_xrfz · 2024-10-30

**Soundness:** 2
**Presentation:** 1
**Contribution:** 2
**Rating:** 3
**Confidence:** 4

**Summary:**

The paper studies the problem of task solving with LLMs. The author proposed EC-Agent which include a reasoning stage and a summary stage. The author tested the proposed method on 4 different datasets, and showed that they are better than previous baselines like CoT and SPP.

**Strengths:**

The paper studies the problem of task-solving which is definitely important. The proposed idea of replacing multi-agent system with a single-agent one is interesting.

**Weaknesses:**

1. The clarity of the paper needs great improvement. The method section is not giving enough details on how each stage is done, and even the prompt and examples provided in the appendix does not seems to match what is proposed in the paper. While the high-level idea seems interesting, the low-level details are completely missing, and the authors seems trying to cover this by introducing many notations that also seems unnecessary. Examples should be provided in the main paper to give concrete idea on how each step is done, or prompt should be included. The paper is also not self-contained as the tasks used is not well-described to even get a high-level understanding of the task.
2. The name of EC-Agent seems very confusing, and the scope of the paper is not clear enough. While task-solving is a very broad problem, I cannot further identify whether the authors are trying to address the problem in reasoning-heave tasks, which they probably is because they explicitly have the reasoning stage included, or they are studying the general problem.
3. The experiment design is quite confusing. This is on one hand related to the scope of the problem to be unclear, and on the other hand the baselines selected. CoT is designed for reasoning problems, while tasks that does not focus on reasoning like Trivia Creative Writing and Codename Narratives are included in the experiments.

**Questions:**

No specific questions. Please refer to the weakness. Overall, I believe this paper needs a complete rewrite to improve the clarity.

---

### Official Review · Reviewer_YSTM · 2024-11-01

**Soundness:** 2
**Presentation:** 1
**Contribution:** 2
**Rating:** 5
**Confidence:** 5

**Summary:**

The paper proposes to solve reasoning problems by introducing an EC-Agent, i.e., an Explicitly Constrained agent where explicit constraints/rules are provided to the LLM to solve a given reasoning problem. The proposed solution is split into a Reasoning Stage, where the LLM is expected to analyze the task type and rules to generate multiple suitable methods and constraints, and finally produce possible solutions to the problem; and a Summary Stage where the LLM acts as an evaluator over the previously generated solutions followed by rectifying the inconsistencies, summarizing, and generating the final answer. The authors show improvements over standard prompting and Chain of Thought over 4 tasks, namely - Trivia Creative Writing, Logic Grid Puzzle, Codenames Collaborative, and Code Generation using both OpenAI and open-sourced LLM models.

**Strengths:**

1) Any reasoning problem requires a model to reason over problem-specific artifacts, take problem-specific rules/constraints into account, and finally generate the answer. As several studies have shown by now, LLMs are not reliable and robust in solving several reasoning problems, and the prompter has to rely on prompt engineering (which may lead to information leakage) and/or LLM's approximate retrieval abilities for outputting the right response. This work proposes to explicitly separate the problem-specific rule-generation component which can later be used by the LLM to generate diverse responses, from which a possibly correct one can be selected, if any. This is surely an approach more sound than directly relying either on the prompter to provide the rules or use prompt engineering techniques.
2) Generating diverse outputs for a given reasoning problem using a noisy generator (LLM in this case) has been shown to be useful too for solving reasoning tasks.
3) The authors show comprehensive experiments over a variety of tasks which include both, natural language and code generation.

**Weaknesses:**

1) Since the proposed system relies on the LLMs' capabilities to generate task-specific rules/constraints, there is no way of verifying the validity of these constraints. The LLM-generated constraints can be unreliable, and thus may lead to cascade of errors when they may be used for generating possible solutions. A reasonable way to avoid this potential issue could be to use a verification module that has access to ground-truth rules/constraints. While this may require a domain expert (human) to provide the rules/constraints, it can potentially reduce hallucinations at the rules/constraints-generation step.
2) It is unclear if and how the proposed method is used for generating constraints using both, explicit and implicit constraints of a given task. If the method is able to derive constraints using the list of task-specific implicit constraints, where are these implicit constraints originally coming from? Then, is it expected for the LLM to have any information about the task before-hand? A detailed example of constraint generation can be extremely useful.
3) Overall, it is hard to understand the intuition behind a lot of technical and engineering decisions taken for the proposed approach in the paper. Specifically, Section 3 is not clearly written and it may be useful to revise the overall flow for an easier understanding of the approach.

**Questions:**

Major points:
1) What method is utilized to generate diverse solution outputs by the LLM? Is temperature the only parameter changed for re-prompting the LLM multiple times? Can the authors elaborate on this and how they are checking if the LLM-generated outputs are indeed semantically diverse and not just syntactically different?
2) Line 188: What is meant by generating `multiple methods'? Is this similar to an algorithm or a reasoning trace (as in CoT) that the LLM is generating?
3) Line 238: What is the multi-sampling approach used in the method, and how does it `enhance robustness'?
4) From the current description of the approach and experiments in Section 3 and 4, my understanding is that the approach is a version of CoT with the additional steps of explicit constraint generation, multiple solution outputs and LLM-based verification. Is this understanding correct? (The question is only to have a better understanding of the approach, and the reviewer acknowledges that the approach is significantly different from the original CoT version.)

Minor points:
1) Line 49: What do the authors mean by `evaluate the task from different perspectives'? This seems to be a vague statement in the context of objective reasoning problems, and elaboration may help understand this better.
2) Line 415: Typo in `observer'

---

### Official Review · Reviewer_95EP · 2024-11-03

**Soundness:** 3
**Presentation:** 2
**Contribution:** 3
**Rating:** 6
**Confidence:** 2

**Summary:**

The paper proposes explicit-constraint agents (EC-agents), as a prompting and scaffolding technique for LLMs to solve specific tasks. In brief, the LLM is prompted to first generate methods and constraints for solving a task and then asked to execute according to each method and constraint it came up with. Finally, it is asked to summarize the takeaways.

The results show performance improvements across multiple tasks and models.

**Strengths:**

As far as I'm aware, this is a novel prompting technique, and the results are encouraging on the tasks it was assessed at.

**Weaknesses:**

The computational cost is only briefly discussed. As far as I understand, the agent is effectively asked to solve the problem multiple times, and then to summarize the takeaway from each. It would have been interesting to see an evaluation of how performance varies with the number of samples, and to have a clearer comparison with the computational cost of alternative techniques like SPP and CoT.

**Questions:**

How does performance scale with number of samples, and how does total compute compare against other methods?

---

### Official Review · Reviewer_RSdz · 2024-11-04

**Soundness:** 2
**Presentation:** 3
**Contribution:** 1
**Rating:** 3
**Confidence:** 4

**Summary:**

The paper presents a prompting strategy claiming to improve reasoning - they do so via a two stage approach of "Reasoning" and "Summarizing" with the expectation that the reasoning step enumerates methods / corresponding constraints which can yield execution outputs. This context is fed (during the summarization phase) the allow for a review section (to adjust reasoning process) and a summary section which generates the final result.

**Strengths:**

The paper is well written, it was easy to understand the core contribution and their experiments.

**Weaknesses:**

I believe that the core contribution of the paper is a prompting strategy which will not stand the test of time as newer models are trained. For instance, its unclear if there is a fundamental reason to why this prompting strategy help? Are there insights from the training set that the authors can provide which support their arguments?

Limited Evaluation :
The paper takes up GPT4 (which seems to be the only "strong" model in their evals) and other weaker models (LLAMA-8b, gemma-9b, mistral-7b). Maybe they can show their main results on diverse models?

Furthermore, the results themselves are not particularly encouraging. (Plots in Appendix C are missing baselines which makes it hard to read). Table 3 only compares to standard prompting. For tables 1 & 2, the largest gap between EC (proposed work) and *any of the baseline* seems to be 2-3% (except Trivia.C.W in table 3 where it is 7%).

Standard prompting performs in the range of 60-80% for the domains taken up for empirical analysis. There are harder planning problems where the performance is extremely poor (0-20% like Travel Planning, PPDL planning etc.). Can the authors discuss why those domains are out of scope of their work, or if they plan to extend their empirical study?

Methodology :

The method seems to assume good instruction following ability (especially at generating valid constraints, methods etc.) followed by self-reflection & summarization. While LLMs summarization is relatively better (except grounding), self-reflection does not generalize to planning / reasoning scenarios, and instruction following (without explicit finetuning) fails simple graph-coloring (a typical "reasoning") problem. Maybe the authors can discuss more about why is it that this prompting strategy should work out of the box?

**Questions:**

See "Weaknesses".

---

### Note · Authors · 2024-11-26

I have read and agree with the venue's withdrawal policy on behalf of myself and my co-authors.